# A new bottom-up method for the standard analysis and comparison of workforce capacity in mental healthcare planning: Demonstration study in the Australian Capital Territory

**Mary Anne Furst**[1]*, **Jose A. Salinas-Perez**[2], **Mencia R. Gutiérrez-Colosia**[3], **Luis Salvador-Carulla**[1]

**1** Centre for Mental Health Research, Australian National University, Canberra, Australian Capital Territory, Australia, **2** Department of Quantitative Methods, Universidad Loyola Andalucia, Sevilla, Spain, **3** Department of Psychology, Universidad Loyola Andalucia, Sevilla, Spain

* Mary.Furst@anu.edu.au

**Data Availability Statement:** The data underlying the results presented in the study are available

## Abstract

The aims of this study are to evaluate and describe mental health workforce and capacity, and to describe the relationship between workforce capacity and patterns of care in local areas. We conducted a comparative demonstration study of the applicability of an internationally validated standardised service classification instrument—the Description and Evaluation of Services and Directories—DESDE-LTC) using the emerging mental health ecosystems research (MHESR) approach. Using DESDE-LTC as the framework, and drawing from international occupation classifications, the workforce was classified according to characteristics including the type of care provided and professional background. Our reference area was the Australian Capital Territory, which we compared with two other urban districts in Australia (Sydney and South East Sydney) and three benchmark international health districts (Helsinki-Uusima (Finland), Verona (Italy) and Gipuzkoa (Spain)). We also compared our data with national level data where available. The Australian and Finnish regions had a larger and more highly skilled workforce than the southern European regions. The pattern of workforce availability and profile varied, even within the same country, at the local level. We found significant differences between regional rates of identified rates of psychiatrists and psychologists, and national averages. Using a standardised classification instrument at the local level, and our occupational groupings, we were able to assess the available workforce and provide information relevant to planners about the actual capacity of the system. Data obtained at local level is critical to providing planners with reliable data to inform their decision making.

from the Metadata Repository (https://rsph.anu.
edu.au/research/projects/glocal-global-and-local-
observation-and-mapping-care-levels/metadata-
repository).

**Funding:** This research is supported by the Bupa
Foundation.

**Competing interests:** The authors have declared
that no competing interests exist.

## Introduction

Ensuring that health systems have sufficient availability and distribution of appropriately
skilled workers is critical if they are to function as intended, and to meet challenges as they
arise. An understanding of current workforce capacity is key to being able to plan for future
workforce requirements in the face of challenges such as the changing health needs of commu-
nities, the lengthy and costly training of health professionals, financial constraints, and pat-
terns of professional migration [1]. Knowledge of the profile and capacity of the health
workforce underpins planning and policy making; promotes accountability and capacity
building at all levels of the health system [2]; and informs monitoring and evaluation of strate-
gies to address issues of staffing adequacy; capacity and distribution; and to quantify projec-
tions of future workforce needs [2–4].

Access to current and accurate workforce data is even more critical in mental healthcare
due to its complexity—the number of different types of care and of professionals across differ-
ent sectors of care—especially as it transitions from a hospital to a balanced model of service
delivery; and the challenges of chronic underfunding, inequity of human resource distribution,
and difficulties with recruitment and retention of staff [2, 4–8]. In crisis situations such as the
Covid-19 pandemic, data on current workforce availability and capacity provides planners
with critical information to allow them to leverage the available mental health workforce to
deliver large scale interventions [9, 10]. The World Health Organization (WHO) has called for
systematic assessment of current staffing in mental health as a prerequisite of evidence
informed policy and planning in service delivery [10], and recommended that workforce eval-
uation be an integral part of Human Resources (HR) policy, planning and training. In Austra-
lia, the recent Productivity Commission report has called for a skilled mental health workforce
that is responsive to local need. It identified a need for more psychiatrists and mental health
nurses, especially in regional and rural areas, a strengthened peer workforce, and the impor-
tance of building on the capacity of the indigenous workforce. These improvements should
include the availability of "standardised and comparable data at all levels" to compare work-
force availability and capacity, and redress inequities in workforce distribution; an under-
standing of workforce characteristics; and integrating workforce strategy with service and
infrastructure planning by aligning key system characteristics such as the availability and loca-
tion of practitioners with consumer need [4].

However, the complexity of the mental health system and of its workforce—skilled in a
range of disciplines and levels of qualification and employed across a range of sectors and
types of services—and with a range of qualification levels—presents huge challenges to obtain-
ing reliable, comprehensive and comparative data. Requirements for registration as a health
professional vary internationally [11]. Additionally, conceptual ambiguity and terminological
variability in mental healthcare limit the gathering of meaningful data. For example, the name
"psychologist" may refer to a registered or a clinical psychologist; psychologists may be regu-
lated as a health professional in the Australian Health Practitioner Regulation Agency
(AHPRA) [12]; or classified as a "social professional" according to the International Standard
Classification of Occupations (ISCO-08) [13]. AHPRA does not include "Psychotherapist" as a
regulated category of health practitioner, but according to ISCO-08, a psychotherapist can
refer to a psychologist, but is included within the 'social and religious professionals' category,
together with sociologists, philosophers and social workers, instead of 'health professionals'. In
the US and Canada, a licensed psychotherapist requires a doctoral degree. A mental health
nurse may or may not have specialist mental health qualifications, and this distinction is not
necessarily documented even in professional registers, such as AHPRA. Umbrella terms such
as "case manager" and "counsellor" describe roles which may be occupied by any of several

different types of professional, each bringing quite different skill sets [14, 15]. It is unclear what types of professional, or of additional training, should be required in emergency departments and other acute general health settings which regularly deal with people with mental illness [16].

On the other hand, non-specialist health workers or lay workers comprise a significant part of the mental health workforce, particularly in Low and Middle Income Countries [17].

Outside the health sector, there is even more confusion, with an array of ill-defined occupations such as community mental health worker, support worker, case worker, key worker and case manager or navigator, positions which may be filled at different times and in different organisations by workers with anything from a six month vocational training certificate through to tertiary trained health or social professionals.

Moreover, available methods for providing data on the delivery of mental healthcare are scarce and subject to serious methodological limitations. This includes lack of standardisation, variable and/or ambiguous terminology, and the risk of ecological fallacy by the use of "top down" national indicators to inform decision making at the local level, where the actual allocation of staff and services takes place. Currently available methods also frequently fail to include information from other sectors providing care to people experiencing mental ill health, and so provide an unbalanced and incomplete picture of care availability [18]. Workforce data at the local level may also be more difficult to obtain due to problems in quality and variability of data. For example, the World Health Organisation(WHO) atlas [19] provides data at national level; and national agencies such as the Australian Institute of Health and Welfare (AIHW) [20] provide state or territory level data. While professional registers record currently qualified and registered health professionals and those in occupations with protected titles, there is no equivalent database of staff working in other roles. Additionally, where classification of occupations is based only on the nature of the work performed, as in ISCO [13] and the European skill/Competences, qualifications and Occupations (ESCO) [21], rather than on the characteristics or background of the person performing, then, as noted in the examples described above, the real capacity and skills of the workforce may not be identified.

A standardised terminology and framework that can be used across different sectors is urgently needed. The WHO "One Health" [22] model, calls for programs, policies, legislation and research in which multiple sectors communicate and work together to achieve better public health outcomes. This "whole systems" approach should encompass all sectors and workers providing care to a defined population group, to provide a comprehensive picture of the profile and capacity of the health workforce in a region, and to be able to compare it with other regions.

In this study, we present a new method to evaluate and describe the mental health workforce profile and capacity in local areas, using a health ecosystems approach [23] together with the Description and Evaluation of Services and DirectoriEs (DESDE) [24], a standardised and internationally validated assessment system. We describe the relationship of the workforce capacity to patterns of regional care provision, and conduct a comparative demonstration study of its applicability in the Australian Capital Territory (ACT), two other urban health districts in Australia (Sydney and South East Sydney) and three benchmark international urban health districts in Europe (Helsinki-Usimaa (or "Helsinki") (Finland), Gipuzkoa (Spain) and Verona (Italy)).

## Method

This is a demonstration study of the use of DESDE-LTC to describe the workforce capacity in local areas following a healthcare ecosystem approach and a whole system perspective of mental healthcare.

## Key models, terms and groupings

We have adapted, wherever possible, the ISCO-08 classification providing a series of modifications and additions to increase the clarity of the definitions for a standard assessment of workforce capacity. "Workforce" is defined as the people engaged in or available for work, either in a country or area or in a particular service. The term "capacity" follows the Talent Management Model [25] in human resource management as "the knowledge and skills, qualifications and entitlement of an individual to conduct a defined set of activities in practice that defines the maximum ability that exists at present in real world conditions". It is characterised by the "power, ability or possibility of doing something or performing") [26]. This concept is different to "capability", which refers to the higher level of ability that could be demonstrated under the right or ideal conditions. Capacity is also different from current performance, as it takes into account the knowledge and skill set of the individual. For example, a nurse working in a case management job would have a capacity different to a health worker, due to his/her professional background.

In order to assess workforce capacity, it is necessary to consider the differences between profession, occupation and job. Profession is mentioned, but not defined, in ISCO-08. A "job" is a set of tasks and duties performed, or meant to be performed, by one person, including for an employer or in self-employment [13]. The job is related to the specific conditions, activities and skills defined in a particular contract. An "occupation" is the set of jobs whose main tasks and duties are characterised by a high degree of similarity. It refers to the kind of work performed in a job that is analogous to the work in other settings and countries.

Therefore, ISCO-08 classifies occupations, and not jobs. Finally, a "profession" is a disciplined group of individuals who adhere to ethical standards. This group positions itself as possessing special knowledge and skills in a widely recognised body of learning derived from research, education and training at a high level, and is recognised by the public as such. It is also prepared to apply this knowledge and exercise these skills in the interest of others [27]. Therefore, professionals are accountable to those served and to society [28].

Professions are recognised by standard education programs and their related qualifications and entitlements at national level [29]. Thus "jobs" have a significant variability across jurisdictions and are difficult to compare, "occupations: show a lower level of heterogeneity, and those occupations that are associated to "professions" have international standards, and are less difficult to compare internationally. In addition, professions are associated to a more homogeneous skill set.

Here we use an adaptation of the occupational groupings developed by the World Health Organisation (WHO) [19], and ISCO-08 which distinguish between health and non-health occupations. When describing the types of services employing staff, we also make a distinction between "core health" services which have an explicit health focus, and which are mainly staffed by clinical health professionals, with 3 years or more of training, and "other", non-clinical services, focused on community participation and promotion of independence, such as skills development, or assistance with accommodation, housing and employment [30]. The latter are staffed predominantly with people with certificate or diploma level training, or without formal training, and to a lesser extent with tertiary professionals such as social workers and occupational therapists.

In our study, we have included all people employed to provide direct support to the target population of each service, according to their professional background. This addresses the issues of ambiguity regarding occupational titles such as "counsellor", "psychotherapist" or "case manager" where the position may be held by a range of different professionals, and provides a more accurate picture of the real capacity of the workforce accordingly. The "clinical

health professionals" group included psychiatrists/registrars, other physicians, psychologists, and nurses; while "allied health professionals" refers to any tertiary qualified allied health professional employed to provide direct care to the target population, such as social workers, or less frequently, occupational therapists. Other non-tertiary qualified occupations encompass a range of roles and responsibilities, but lack standardised occupation titles, responsibilities or prerequisites enabling accurate distinctions between them, and thus are here described collectively as "others". These roles may include, but are not limited to, such titles as "support worker", "mental health worker", "case worker" and "recovery coach". They may include qualifications at diploma level or other grade level with less than three years training, or they may require no specific training.

## Description and Evaluation of care Systems and DirectoriEs for Long Term Care (DESDE-LTC)

DESDE-LTC is an internationally standardised and validated instrument for the standard description and classification of services across different sectors [31], which was used to develop Atlases and Directories of Mental Healthcare of the included regions. Using the DESDE-LTC tool, the atlases provided a holistic view of the local context including socio-economic and socio-demographic data, service availability and both placement (i.e. bed and/or place) and workforce capacity. The DESDE-LTC identifies individual services providing care on a temporally and administratively stable basis, described as Basic Stable Inputs of Care (BSICs); and describes the Main Type of Care they provide (MTCs). MTCs are classified according to sub-categories of one of six main branches of care (Residential, Outpatient, and Day Care). This provides a standardised framework for an analysis of the type of care delivered, the type of worker delivering it, and how the workforce is distributed. Use of DESDE-LTC enables valid comparison across regions and countries despite the different levels of care, units of analysis, and terminology which characterise mental health systems. It is a multiaxial system, classifying services along several axes including their target population; type of care; the sector of care; and the professional providing the service.

## Inclusion criteria

Inclusion criteria was that of the Integrated Atlases as follows:

All staff working in services and providing direct care to adults with a lived experience of mental illness in services in all sectors were included. Staff not providing direct care, such as administrative staff, were not included. The workforce was that employed in services which met the following criteria:

1. **The service targets adults with a lived experience of mental illness**: The primary reason for using the service is a mental health issue or a psychosocial disability including any diagnosis of mental disorders (ICD-10, section F).

2. **The service is universally accessible**: The study focuses on services that are at least in part universally accessible, regardless of if they are publicly or privately funded. The inclusion of services requiring significant out of pocket payment and fully private insurance would give a misleading picture of the resources available to most people living with mental illness and obscures the data for evidence-informed planning of the public health system. These services should be mapped in a separate layer of information.

3. **The service is within the boundaries of the study region**: The inclusion of services that are within the boundaries of the study region is essential to have a clear picture of the local availability of resources.

4. **The service provides direct care or support to consumers**. This excludes services which may coordinate other services, but which do not have direct contact with consumers.

5. Services that do not have guaranteed funding for three years receive an extension code "v" to differentiate them from stable services and to facilitate the description of the robustness of the system.

## Catchment areas

We have mapped the workforce profile and capacity of three urban regions under the jurisdiction of regional health bodies in Australia: Australian Capital Territory Primary Health Network (ACT PHN), Sydney Local Health District (SLHD) and South East Sydney Local Health District (SESLHD), as well as three urban health districts in Europe: Helsinki- Uusimaa (Finland), Gipuzkoa (Spain); and Verona (Italy). The adult population of these regions ranged from 277,019 in the ACT to 1,206,446 in Helsinki-Uusimaa.

In Australia, healthcare, including mental healthcare, is coordinated at the regional level by a network of 31 Primary Health Networks, which are broadly similar geographically to Local Health Districts, and are responsible for management of public hospitals and community mental health centres. The Helsinki-Uusimaa study area is comprised of 26 municipalities and five geographic sub areas including the capital of Finland. Helsinki-Uusimaa has been regarded as a key area for demonstration studies of general healthcare [32] and mental healthcare [33, 34]. It includes eight public psychiatric hospitals. Primary mental healthcare is provided at health centres and secondary and tertiary care by the hospital district of Helsinki-Uusimaa. Verona is a province of the Veneto region in Italy. Mental healthcare in Italy is based on a community model organised into local health districts based on geographic area, each with its own Department of Mental Health providing a range of inpatient and outpatient services. The mental health provision and coding was conducted as part of the REFINEMENT study [33]. The mental health system in Spain also follows a community model organised in catchment areas with full devolution of funding and management to the regions [35]. San Sebastian is the capital of the Gipuzkoa province of the Basque country region in Spain. Gipuzkoa's mental health network includes 13 catchment areas, each with its own mental health centre [36]. All countries in the study are classified as high income countries [37], with Australia having the highest GDP per capita, followed by Finland [38].

## Procedure

Data from the ACT reference area was collected by researchers at The Australian National University (ANU) and the University of Sydney with DESDE-LTC in 2016 as part of the Integrated Atlas of Mental Health of the Australian Capital Territory Primary Health Network region [39]. Ethics approval for the ACT data was granted by ACT Health Research Ethics and Governance Office (protocol ETHLR.16.094).

This data was compared to data registered in the metadata repository on service provision and workforce capacity of the GLOCAL Project (Global and Local Observation and mapping of CAre Levels) that synthesises information from all published studies using the ESMS/DESDE system in the world. (The European Service Mapping Schedule (ESMS) is an earlier version of DESDE). In this case, information from ACT was compared with data from other

areas in Australia and the world where the data gathering and quality was supervised by members of our team (LSC and MGC). This included information available in the repository from the other two Australian areas: South Eastern Sydney Local Health District, and Sydney Local Health District [40]. Verona and Helsinki resource data came from the service mapping carried out in the REFINEMENT project (REsearch on FINancing systems' Effect on the quality of MENTal healthcare) [41]: eight areas from eight countries were mapped in 2013 by using the DESDE-LTC coding system [30, 33]. Finally, data from the Gipuzkoa area were collected from the Mental Health Atlas developed by the Psicost Research Association in 2013 [36] (updated to 2015), also using the DESDE-LTC tool.

Data showing the rate of psychiatrists and psychologists at national level was obtained from the WHO Global Health Observatory [42].

## Data analysis

Workforce numbers were calculated as Full Time Equivalents (FTEs) per 100,000 adult population (aged 18 years and over) and analysed according to: (i) occupation; (ii) to their representation in the main branches of care in the DESDE-LTC instrument; and (iii) in relation to service availability (the number of workers in relation to the number of services of each type of care (MTC) per 100,000 population), which provides an average figure for the size of teams. It is important to use FTEs as the unit of measurement so that the data is not distorted by counting part-time and casual staff. Australian regions included data from all main branches of care: however, from the international regions, data in the smaller branches of Accessibility, Information or Self help main branches were not available. In addition, workforce profile was studied through percentages over the overall professionals in each type of care and in each health area.

## Results

Table 1 shows detailed rates of professionals according to main type of care, and Figs 1–3 show workforce composition.

Helsinki's overall workforce rate was the highest of all regions in the study, at 247.97 staff per 100,000 adults, and SESLHD the lowest, with 123.74 staff per 100,000 adults. ACT had the second highest workforce rate (Table 1).

### Professional groups availability and workforce composition

**Psychiatrists.**   The rate of psychiatrists in ACT (10.83 per 100,000 adults) was within the range of the other Australian regions (7.99 and 15.97 per 100,000 adults in SES and SLHD respectively) (Table 1), although their distribution differed, with fewer employed in acute inpatient wards in ACT, and more psychiatrists working outside the hospital setting. In international comparison, ACT had the lowest rate of psychiatrists, with Verona (20.23 per 100,000 adults) and Helsinki (24.01 per 100,000 adults) providing close to double the rate of psychiatrists available in ACT and Gipuzkoa (11.57) (Table 1). Helsinki's rate of psychiatrists was the highest of all regions (Figs 1–3), particularly in sub-acute residential care (Table 1; Fig 1). In outpatient care, the rate of psychiatrists in Helsinki (14.37 per 100,000 adults) (Table 1) was more than double all other regions (Table 1; Fig 2).

The total rate of psychiatrists in SES (7.99 per 100,000), Verona (20.23) and Helsinki- Uusimaa (24.01)(Table 1) was significantly different to their respective national averages (13.53. 5.98 and 48.04). Psychiatrists comprised 13% of the total workforce in Verona, 10.3% in SLHD, 9.7% in Helsinki, 7.9% in Gipuzkoa, 6.5% in SESLHD and 6.1% in ACT (Figs 4–6). They comprised a smaller part of the acute residential care workforce in the ACT (6.4%), than

**Table 1. Full time equivalents: Comparison between areas according to type of care and type of professional.**

| | | Psychiatrists/ registrars | Other physicians | Psych ologists | Nurses | Assistant nurses | Social workers | Occupational therapists | Others | Total |
|---|---|---|---|---|---|---|---|---|---|---|
| **Acute ward** | Gipuzkoa (Spain) | 3.28 | 0.00 | 1.09 | 3.75 | 4.84 | 0.00 | 0.31 | 0.47 | 13.74 |
| | Verona (Italy) | 4.17 | 0.00 | 1.27 | 15.94 | 0.00 | 0.76 | 0.00 | 6.11 | 28.25 |
| | Helsinki (Finland) | 3.24 | 0.00 | 1.48 | 19.31 | 0.00 | 1.66 | 0.99 | 11.40 | 38.08 |
| | Sydney LHD | 7.75 | 0.00 | 1.09 | 54.14 | 0.00 | 3.19 | 2.25 | 0.22 | 68.64 |
| | SE Sydney LHD | 4.34 | 0.00 | 0.39 | 39.31 | 0.00 | 1.38 | 0.62 | 2.08 | 48.12 |
| | ACT | 2.89 | 0.00 | 2.09 | 36.24 | 0.00 | 1.44 | 1.26 | 1.14 | 45.06 |
| **Non-Acute 24h physician (e.g. subacute ward, crisis home)** | Gipuzkoa (Spain) | 1.29 | 1.27 | 1.38 | 4.60 | 19.26 | 1.02 | 0.48 | 13.20 | 42.50 |
| | Verona (Italy) | 2.33 | 0.00 | 1.26 | 8.19 | 0.00 | 0.20 | 0.00 | 20.06 | 32.04 |
| | Helsinki (Finland) | 5.41 | 0.08 | 1.68 | 32.49 | 0.00 | 2.38 | 1.76 | 23.83 | 67.63 |
| | Sydney LHD | 0.98 | 0.00 | 0.25 | 9.25 | 0.00 | 0.94 | 0.67 | 0.00 | 12.09 |
| | SE Sydney LHD | 0.45 | 0.00 | 0.31 | 5.54 | 0.00 | 0.31 | 0.46 | 0.08 | 7.15 |
| | ACT | 1.08 | 0.00 | 0.72 | 14.71 | 0.00 | 0.36 | 0.36 | 1.26 | 18.49 |
| **Community residential** | Gipuzkoa (Spain) | 0.02 | 0.07 | 0.70 | 0.13 | 1.20 | 0.68 | 0.56 | 21.67 | 25.03 |
| | Verona (Italy) | 1.71 | 0.76 | 2.11 | 2.42 | 0.00 | 0.90 | 0.00 | 29.74 | 37.64 |
| | Helsinki (Finland) | 0.33 | 0.07 | 0.00 | 9.28 | 0.00 | 0.43 | 0.27 | 45.21 | 55.59 |
| | Sydney LHD | 0.00 | 0.00 | 0.00 | 0.00 | 0.00 | 0.00 | 0.00 | 2.32 | 2.32 |
| | SE Sydney LHD | 0.00 | 0.00 | 0.00 | 0.00 | 0.00 | 0.00 | 0.00 | 1.39 | 1.39 |
| | ACT | 0.72 | 0.00 | 0.29 | 9.85 | 0.00 | 0.36 | 0.58 | 14.58 | 26.38 |
| **Outpatient health** | Gipuzkoa (Spain) | 6.79 | 0 | 2.82 | 3.8 | 0 | 1.75 | 0 | 0 | 15.16 |
| | Verona (Italy) | 7.88 | 0 | 1.47 | 7.26 | 0 | 0.96 | 0 | 7.9 | 25.47 |
| | Helsinki (Finland) | 14.37 | 0.1 | 7.78 | 32.91 | 0 | 4.73 | 3 | 7.13 | 70.02 |
| | Sydney LHD | 7.22 | 0.27 | 2.86 | 19.96 | 0 | 4.03 | 1.34 | 4.23 | 39.91 |
| | SE Sydney LHD | 3.14 | 0 | 5.91 | 8.22 | 0 | 3.47 | 1.01 | 15.74 | 37.49 |
| | ACT | 6.1 | 0 | 15.88 | 25.68 | 0 | 7.37 | 3.61 | 9.24 | 67.88 |
| **Outpatient Social** | Gipuzkoa (Spain) | 0 | 0 | 0 | 0 | 0 | 0 | 0 | 0 | 0.00 |
| | Verona (Italy) | 0 | 0 | 0 | 0 | 0 | 0 | 0 | 0 | 0.00 |
| | Helsinki (Finland) | 0 | 0 | 0.17 | 0.17 | 0 | 0.41 | 0 | 0.33 | 0.91 |
| | Sydney LHD | 0 | 0 | 0 | 0 | 0 | 0 | 0 | 29.43 | 29.43 |
| | SE Sydney LHD | 0.06 | 0 | 0.31 | 0.78 | 0 | 0.62 | 0.77 | 22.36 | 24.90 |
| | ACT | 0 | 0 | 0 | 0.14 | 0 | 0.72 | 0 | 14.54 | 15.40 |

(*Continued*)

**Table 1.** (Continued)

| | | Psychiatrists/ registrars | Other physicians | Psych ologists | Nurses | Assistant nurses | Social workers | Occupational therapists | Others | Total |
|---|---|---|---|---|---|---|---|---|---|---|
| **Day care health** | Gipuzkoa (Spain) | 0.19 | 0 | 0.73 | 0.6 | 0.03 | 0.54 | 1.22 | 9.31 | 12.62 |
| | Verona (Italy) | 4.14 | 0 | 1.19 | 12.94 | 0 | 1.89 | 0 | 12.42 | 32.58 |
| | Helsinki (Finland) | 0.66 | 0 | 0.58 | 3.81 | 0 | 0.56 | 0.83 | 2.36 | 8.80 |
| | Sydney LHD | 0.02 | 0 | 0.22 | 0.07 | 0 | 0 | 0.11 | 0.31 | 0.73 |
| | SE Sydney LHD | 0 | 0 | 0 | 0 | 0 | 0 | 0 | 0 | 0.00 |
| | ACT | 0.04 | 0 | 0.55 | 0.36 | 0 | 0 | 0.36 | 0 | 1.31 |
| **Day care work/work related** | Gipuzkoa (Spain) | 0 | 0 | 0 | 0 | 0 | 0 | 0 | 33.88 | 33.88 |
| | Verona (Italy) | 0 | 0 | 0 | 0 | 0 | 0 | 0 | 0 | 0.00 |
| | Helsinki (Finland) | 0 | 0 | 0 | 0.04 | 0 | 0 | 0.17 | 6.1 | 6.31 |
| | Sydney LHD | 0 | 0 | 0 | 0 | 0 | 0 | 0 | 1.56 | 1.56 |
| | SE Sydney LHD | 0 | 0 | 0 | 0 | 0 | 0 | 0 | 0.27 | 0.27 |
| | ACT | 0 | 0 | 0 | 0 | 0 | 0 | 0 | 0 | 0.00 |
| **Day care other** | Gipuzkoa (Spain) | 0 | 0 | 0.71 | 0.03 | 0 | 0.01 | 0 | 2.97 | 3.72 |
| | Verona (Italy) | 0 | 0 | 0 | 0 | 0 | 0 | 0 | 0 | 0.00 |
| | Helsinki (Finland) | 0 | 0 | 0 | 0 | 0 | 0 | 0 | 0.46 | 0.46 |
| | Sydney LHD | 0 | 0 | 0 | 0 | 0 | 0 | 0 | 0 | 0.00 |
| | SE Sydney LHD | 0 | 0 | 0 | 0 | 0 | 0 | 0 | 4.42 | 4.42 |
| | ACT | 0 | 0 | 0 | 0 | 0 | 0 | 0 | 2.63 | 2.63 |
| **Residential Total** | Gipuzkoa (Spain) | 4.59 | 1.34 | 3.17 | 8.48 | 25.30 | 1.70 | 1.35 | 35.34 | 81.27 |
| | Verona (Italy) | 8.21 | 0.76 | 4.64 | 26.55 | 0.00 | 1.86 | 0.00 | 55.91 | 97.93 |
| | Helsinki (Finland) | 8.98 | 0.15 | 3.16 | 61.08 | 0.00 | 4.47 | 3.02 | 80.44 | 161.30 |
| | Sydney LHD | 8.73 | 0.00 | 1.34 | 63.39 | 0.00 | 4.13 | 2.92 | 2.54 | 83.05 |
| | SE Sydney LHD | 4.79 | 0.00 | 0.70 | 44.85 | 0.00 | 1.69 | 1.08 | 3.55 | 56.66 |
| | ACT | 4.69 | 0.00 | 3.10 | 60.80 | 0.00 | 2.16 | 2.20 | 16.98 | 89.93 |
| **Outpatient Total** | Gipuzkoa (Spain) | 6.79 | 0 | 2.82 | 3.8 | 0 | 1.75 | 0 | 0 | 15.16 |
| | Verona (Italy) | 7.88 | 0 | 1.47 | 7.26 | 0 | 0.96 | 0 | 7.9 | 25.47 |
| | Helsinki (Finland) | 14.37 | 0.1 | 7.95 | 33.08 | 0 | 5.14 | 3 | 7.46 | 70.93 |
| | Sydney LHD | 7.22 | 0.27 | 2.86 | 19.96 | 0 | 4.03 | 1.34 | 33.66 | 69.34 |
| | SE Sydney LHD | 3.2 | 0 | 6.22 | 9 | 0 | 4.09 | 1.78 | 38.1 | 62.39 |
| | ACT | 6.1 | 0 | 15.88 | 25.82 | 0 | 8.09 | 3.61 | 23.78 | 83.28 |

(*Continued*)

**Table 1.** (Continued)

|  |  | Psychiatrists/ registrars | Other physicians | Psych ologists | Nurses | Assistant nurses | Social workers | Occupational therapists | Others | Total |
|---|---|---|---|---|---|---|---|---|---|---|
| **Day Care Total** | Gipuzkoa (Spain) | 0.19 | 0 | 1.44 | 0.63 | 0.03 | 0.55 | 1.22 | 46.16 | 50.22 |
|  | Verona (Italy) | 4.14 | 0 | 1.19 | 12.94 | 0 | 1.89 | 0 | 12.42 | 32.58 |
|  | Helsinki (Finland) | 0.66 | 0 | 0.58 | 3.85 | 0 | 0.56 | 1 | 8.92 | 15.57 |
|  | Sydney LHD | 0.02 | 0 | 0.22 | 0.07 | 0 | 0 | 0.11 | 1.87 | 2.29 |
|  | SE Sydney LHD | 0 | 0 | 0 | 0 | 0 | 0 | 0 | 4.69 | 4.69 |
|  | ACT | 0.04 | 0 | 0.55 | 0.36 | 0 | 0 | 0.36 | 2.63 | 3.94 |
| **Total** | Gipuzkoa (Spain) | 11.57 | 1.34 | 7.43 | 12.91 | 25.33 | 4.00 | 2.57 | 81.50 | 146.65 |
|  | Verona (Italy) | 20.23 | 0.76 | 7.30 | 46.75 | 0.00 | 4.71 | 0.00 | 76.23 | 155.98 |
|  | Helsinki (Finland) | 24.01 | 0.25 | 11.69 | 98.01 | 0.00 | 10.17 | 7.02 | 96.82 | 247.97 |
|  | Sydney LHD | 15.97 | 0.27 | 4.42 | 83.42 | 0.00 | 8.22 | 4.37 | 38.07 | 154.74 |
|  | SE Sydney LHD | 7.99 | 0.00 | 6.92 | 53.85 | 0.00 | 5.78 | 2.86 | 46.34 | 123.74 |
|  | ACT | 10.83 | 0.00 | 19.53 | 86.98 | 0.00 | 10.25 | 6.17 | 43.39 | 177.15 |
| **National** | Italy | 5.98 |  | 3.8 |  |  |  |  |  |  |
|  | Finland | 48.04 |  | 73.52 |  |  |  |  |  |  |
|  | Australia | 13.53 |  | 103 |  |  |  |  |  |  |

in all other regions in the study. In community residential care, clinical professionals overall (psychiatrists, nurses, psychologists) comprised a much higher proportion of the workforce in the ACT (41.2%) than in all other regions, in which the proportion of clinical professionals ranged from zero in the two Sydney regions to 18.6% in Verona (Fig 4).

**Psychologists.** ACT had the highest rate of psychologists of all regions (19.53 per 100,000 adults), with more than double the rate of all regions except Helsinki (11.69 per 100,000 adults)(Table 1). This was particularly the case in outpatient care, where ACT provided 15.88 psychologists per 100,000 adults, compared to SLHD, which had the lowest rate of psychologists of the Australian regions (2.86); and Verona, which had the lowest rate of all regions in the study, with 1.47 psychologists per 100,000 adults working in outpatient care (Table 1; Fig 2).

In residential care, ACT's rate of 3.1 psychologists per 100,000 adults was higher than the other Australian regions, but closer to those of the international regions: Gipuzkoa with 3.17, Verona with 4.64, and Helsinki 3.16 psychologists per 100,000 adults (Table 1). In day care, the picture was slightly different: while ACT again provided more psychologists than the other Australian regions, its rate of psychologists (0.55 per 100,000 adults) was similar to that of Helsinki (0.58), lower than Verona (1.19), and only a third of the rate of psychologists in Gipuzkoa (1.44 psychologists per 100,000 adults)(Table 1; Fig 3).

The total rate of psychologists per 100,000 adult population in all regions (ACT: 19.53, SES: 6.92,SLHD: 4.42, Gipuzkoa: 7.43, Verona: 7.3 and Helsinki: 11.69) (Table 1) was significantly different to their respective national averages (Australia:103; Italy:3.8; Norway:73.52).

National data was not available for Spain.

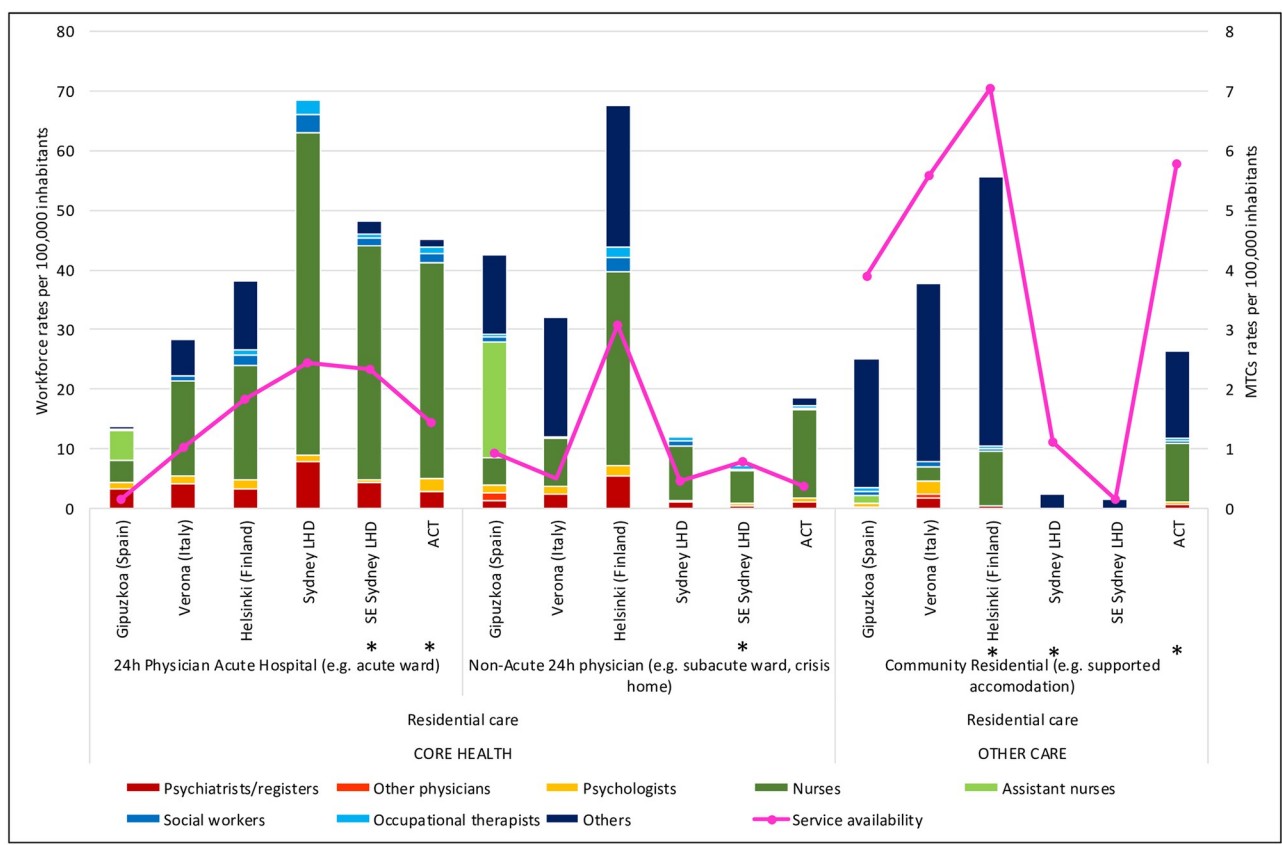

**Fig 1. Workforce rates and availability in adult mental health services for residential care in the six study areas.** *Missing data.

Psychologists comprised 11% of the workforce in ACT compared to 6% in SESLHD, 5% in Helsinki and Verona, 4% in Gipuzkoa and 3% in SLHD (Figs 4–6). ACT had a higher proportion of psychologists than the other Australian regions in health related outpatient care, and this difference was even more pronounced when comparing ACT to the international regions: ACT (15.88 per 100,000 adults) being roughly double that of Helsinki with the next highest rate (7.78), and more than ten times the rate of Verona, which at 1.47 psychologists per 100,000 adults in health related outpatient care had the lowest rate in the study(Table 1; Fig 5).

**Nurses.** ACT provided the highest rate of nurses of the Australian regions, and the second highest in international comparison, after Helsinki. The rate of nurses in all Australian regions (86.98, 83.42 and 53.85 and 83.42 per 100,000 adults in ACT, SLHD and SESLHD respectively) was similar to that of Helsinki (98.01) but significantly more than Gipuzkoa (12.91) and Verona (46.75)(Table 1). Nurses were employed predominantly in residential care in all regions, with Helsinki and ACT providing the highest rates of nurses in community residential (Table 1; Fig 1) and outpatient (Table 1; Fig 2) care. Gipuzkoa alone provided "assistant nurses" in residential (Table 1; Fig 1) and day care (Table 1; Fig 3).

Nurses comprised the largest workforce group in ACT (49%), in the other Australian regions (53.9% in SLHD and 43.5% in SESLHD) and in Helsinki (40%) (Figs 4–6). They were a proportionately higher group in both acute and non-acute hospital residential care in Australian regions than in the international regions (Fig 4). Nurses comprised only 8.8% of the workforce in Gipuzkoa, which was also the only region to provide "Assistant nurses" (17.3% of total workforce).

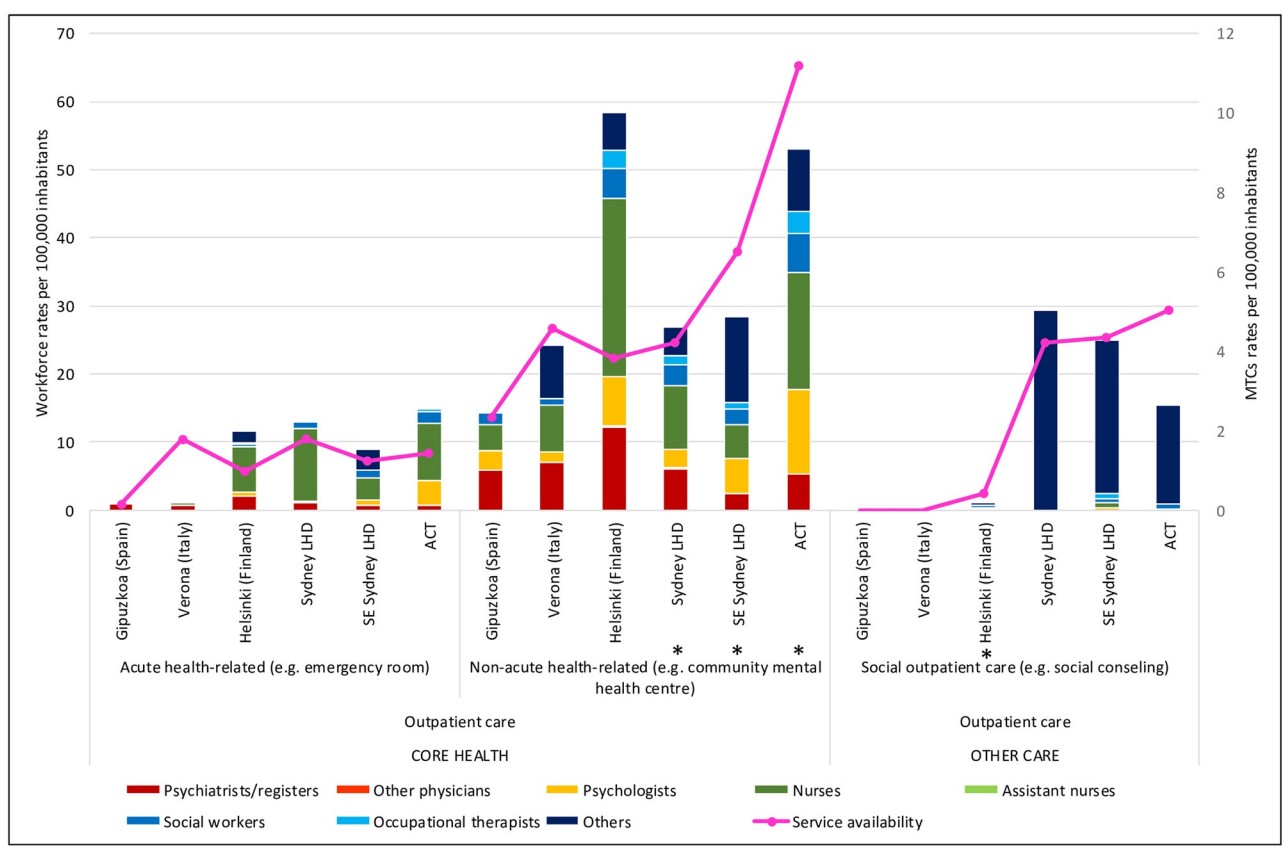

**Fig 2. Workforce rates and availability in adult mental health services for outpatient care in the six study areas.** *Missing data.

**Allied health (social workers and occupational therapists).**    Here again, Helsinki and two of the Australian regions provided similar rates of staff (ranging from 12.59 per 100,000 adults in SLHD to 17.19 in Helsinki) which were significantly higher than those of SESLHD, Gipuzkoa and Verona, which ranged between 4.71 (Verona) and 8.64 (SESLHD)(Table 1). ACT had the second most allied health professionals overall after Helsinki, although in day care, all international regions provided more allied health professionals than all Australian regions (Table 1; Fig 3).

Allied health professionals comprised 9.26% of ACT's workforce, 8.13% in SLHD and 6.98% in SESLHD, compared to 4.48% in Gipuzkoa, 6.93% in Helsinki and 3% in Verona (Figs 4–6).

**Other (non "core health" workforce).**    ACT and the other Australian regions provided significantly lower rates overall of this type of worker than the international regions, with Helsinki providing the highest rate at 96.82 per 100,000 people, and SLHD the lowest at38.07 (Table 1). In residential care, although ACT provided a higher rate of non-core health staff than the other Australian regions (2.54 and 3.55 in SLHD and SESLHD), its rate at 16.98 per 100,000 adults was less than a quarter of that of Helsinki (80.44) and less than half of Gipuzkoa (35.34), and Verona (55.91) (Table 1; Fig 1). However, in outpatient care, ACT provided the lowest rate of non-healthcare staff of the Australian regions (23.78 per 100,000 adults compared to 33.66 in SLHD and 38.1 in SESLHD); although significantly more than Gipuzkoa (0), Helsinki (7.46) or Verona (7.9) (Table 1; Fig 2). These "Other" or non-core health professionals were proportionately smaller in the Australian regions (ACT:24.49% of total workforce;

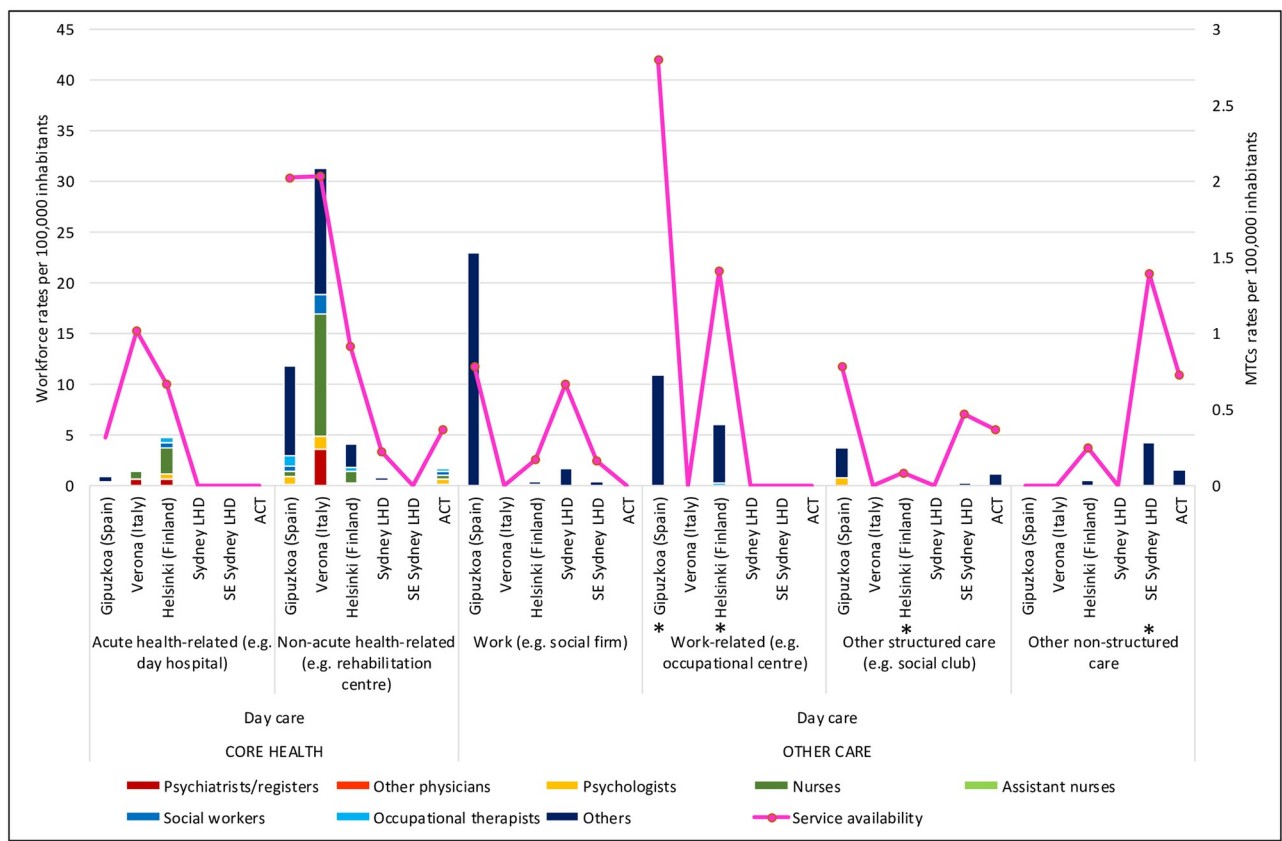

**Fig 3. Workforce rates and availability in adult mental health services for day care in the six study areas.** *Missing data.

SLHD:24.6%; SESLHD: 37.45%) than in the European regions (39.05% in Helsinki, 48.87% in Verona and 55.47% in Gipuzkoa) (Figs 4–6).

## Analysis of workforce in relation to service availability (size of care teams) (Figs 1–3)

Figs 1–3 show the size of the workforce overall in relation to the number of services available, according to the main types of care. In acute residential care, SLHD had the highest workforce rate in relation to rate of service availability of all the regions. While ACT and Helsinki had similar availability of acute residential services, the ACT had a larger workforce capacity. Although the ACT had a higher rate of community residential services than of acute inpatient care, its workforce capacity in community residential services was lower than that of inpatient care (Fig 1).

In outpatient care, workforce rates largely aligned consistently with service availability rates across study regions, with the exception of non-acute health related care in the community, where Helsinki's workforce was significantly greater in relation to its rate of services available than all other regions. ACT's workforce was similar to that of Helsinki, although distributed by a significantly higher rate of services (Fig 2).

Workforce rates similarly largely followed service availability in day care, again with a notable exception in work related daycare, where, although service availability rates in employment support in Gipuzkoa and SLHD were similar, workforce capacity in the Spanish region was far

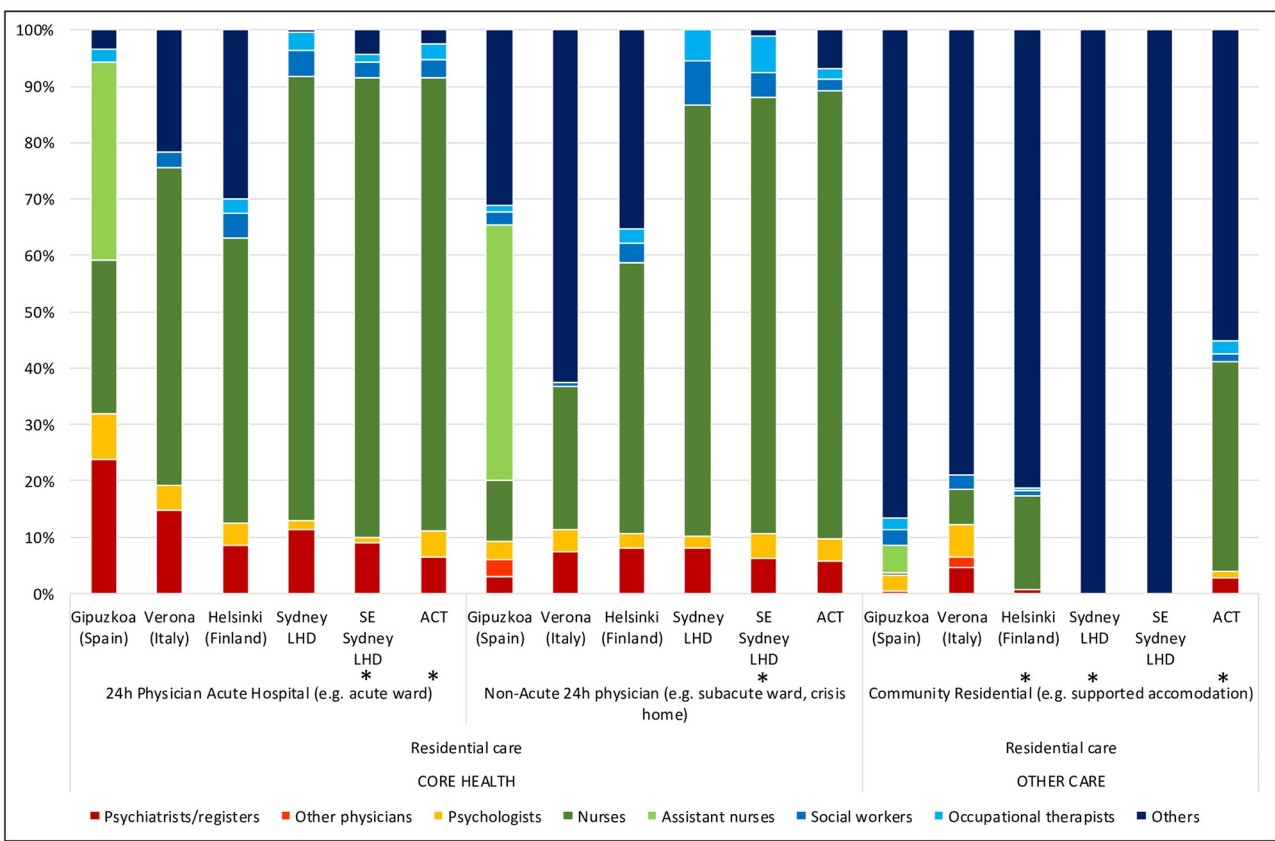

**Fig 4. Percentage distribution of workforce in adult mental health services for residential care in the six study areas.** *Missing data.

greater than that in Sydney. In acute heath related daycare, Verona provided more services than Helsinki, but with a greater workforce capacity (Fig 3).

## Discussion

Traditionally, comparisons on workforce capacity have been carried out using national data gathered by the government, and compiled by international organisations such as OECD or WHO. This information has been used to produce league tables and benchmarking based on crude workforce capacity, and other macro-indicators such as bed availability. This has been considered as "evidence" and applied to health planning [43]. The use of similar global figures of bed availability for ranking and benchmarking Australia's hospital psychiatric beds in comparison to other countries has caused a major international debate [44, 45]. The final agreement was "that finding robust and comparable national data remains very challenging [. . .] with quality of data often a big concern [45]. As is also the case with the total number of beds, the total number of psychiatrists, nurses or psychologists at the macro level cannot be considered "evidence", and even less be used for resource allocation and planning at the local level. These figures are unreliable indicators for mental health planning, due to non- commensurability bias, terminological variability, ecological fallacy, risk of surrogation and objectification [46]. For example, our results showed significant differences in the data at local level from that provided the national level by WHO. Additionally, figures were only available for psychiatrists and psychologists, who in our study comprised less than 40% of the workforce in all types of care. In addition, these sources of information are not reliable. In a recent study by one of

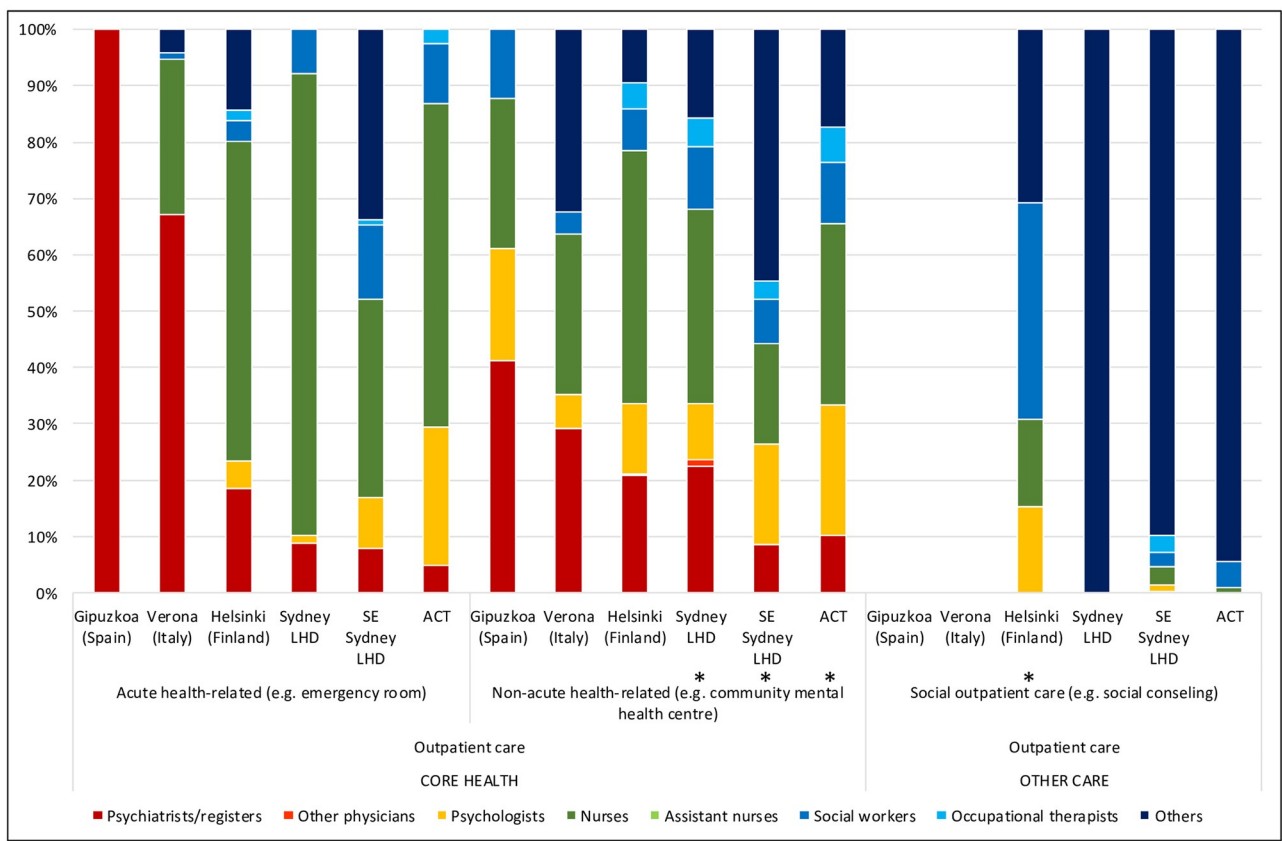

**Fig 5. Percentage distribution of workforce in adult mental health services for outpatient care in the six study areas.** *Missing data.

these groups, over 40% of the policy reports contained errors in accuracy [47]. In the analysis of bed availability previously mentioned, WHO was contacted by the authors after the publication of their study, and the WHO officers questioned the quality of their own published data [45]. Data provided as number of professionals per 100,000 population, and not as FTEs, does not provide the true capacity: the very large difference between national rates of overall number of psychologists identified in WHO data compared to our regional rate of psychologists which were counted according to FTE, suggests that a significant proportion of psychologists could be working more limited hours, and thus less available than the national data suggests.

Under these circumstances, a standard and precise method that allows comparability of data, and where national standards can be disaggregated to the local level, is of outmost importance for international comparison gap analysis; analysis of equality in health provision; for feeding reliable information into the models and decision support system; and finally to guide mental health policy, prioritisation and resource allocation at the local level.

The DESDE-LTC taxonomy provides a framework upon which the pattern of care provision in the mental healthcare system is revealed, using the professional characteristics of the members of care teams working in the service or meso level of the care delivery system. This is the level at which service users engage directly with the system, according to the matrix system of care delivery, which organises the elements of care according to stages of the process and levels of care delivery [48]. Our unit of analysis, the individual service or BSIC, represents the smallest unit of analysis for care delivery. These have also been described as "clinical microsystems": and act as "building blocks" of the healthcare system [49], with common clinical and

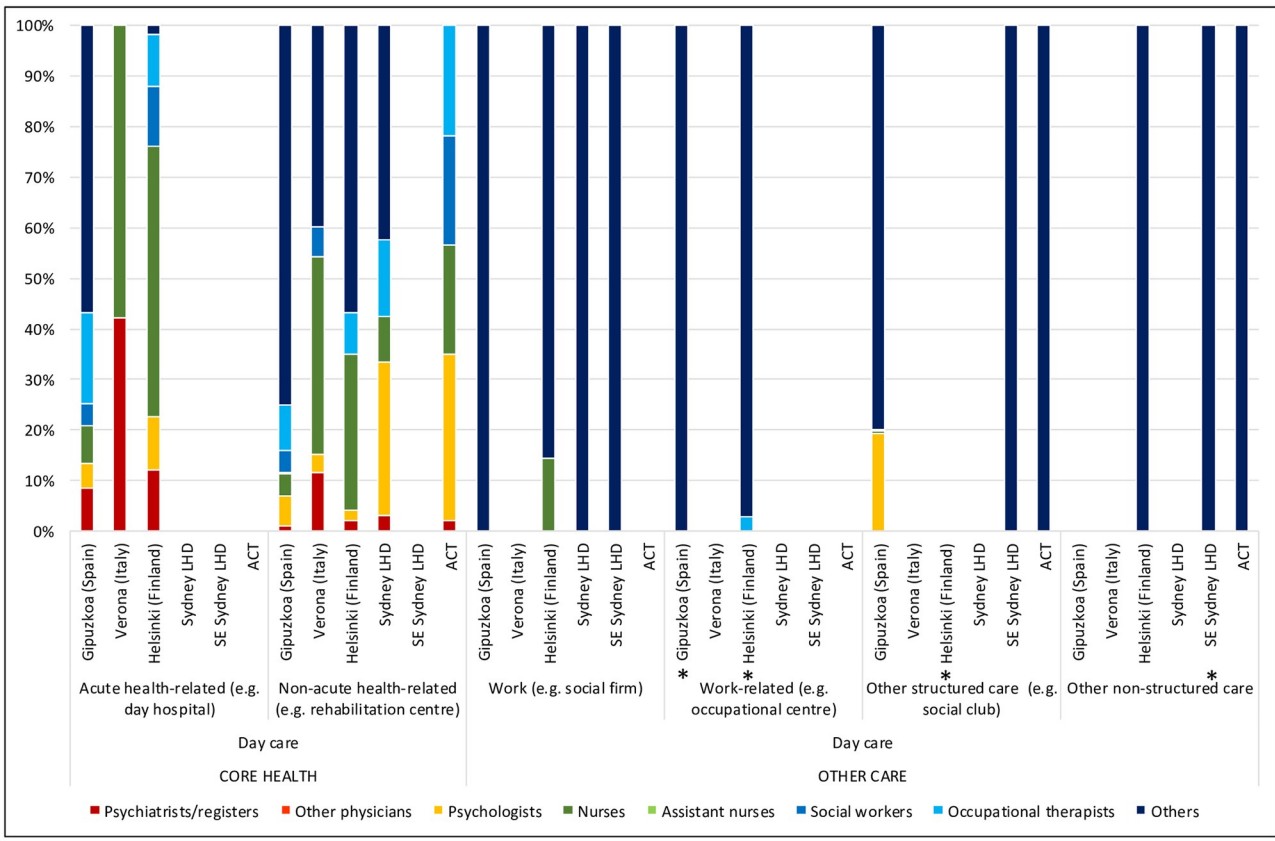

**Fig 6. Percentage distribution of workforce in adult mental health services for day care in the six study areas.** *Missing data.

business aims, linked processes, shared information environment and producing common performance outcomes [50]; as well as providing "the logical locus for linkage between vision and delivery and therefore the "agent for change" within a macrosystem" [51]. Understanding the make-up of these services therefore—who is providing the care within them and what care they are providing—is crucial to understanding the capacity and organisation of the overall workforce.

The results of this study show that the mental health workforce in the ACT was, overall, larger and more highly skilled than all the comparator regions except Helsinki, with the exception of its psychiatrist workforce, the rate of which was higher in the international and one Australian regions, particularly in outpatient care. The high rates of psychologists, but relatively low rates of psychiatrists in the ACT aligns with an observed national trend in Australia [20, 52] of expansion of the psychologist workforce, which is more likely to be individual practitioner based; and contraction of the psychiatrist and psychiatric nursing workforce, which are more likely to provide "more complex, team-based care". This trend may not be providing better outcomes for patients [53] or a more integrated system of care. On the other hand, the ACT had a higher rate of nurses than the other Australian regions.

Mental health nurses have been identified as an area of acute and growing shortage in Australia [3] and their skills may be under-utilised in the delivery of mental healthcare [54] (and possibly unrecognised). This may be particularly so in Spain, where the lower skilled category of "Assistant nurses" comprises a significant part of the "nursing" workforce.

Hospital care, both acute and non-acute, was more likely to be delivered by nurses in Australia than internationally, where some of the care is delivered by lower skilled staff. In the community, however, the difference was less marked, with more nurses in Finland than the ACT in outpatient care, and in Italy in day care.

The community based workforce was larger in the ACT than in the other Australian regions, but they were more likely to be working in health-related than social care services, with few working in outpatient social care, and none at all engaged in work or work related services. When compared to the international regions however, the ACT workforce was less community orientated, with the lowest rates of staff in community residential care and day care. In outpatient care, however, ACT had significantly higher rates of staff than the Spanish and Italian regions. This pattern of workforce distribution is consistent with the pattern of service availability in ACT and other Australian regions, and again reveals a pattern of care provision more focused on individual interactions with service providers, particularly health related providers such as psychologists, than is the case internationally. While this could be consistent with principles of person centred care [55], it also indicates a system more reliant on health based and sessional rather than social type care, which is usually more available in day services. Services which provide opportunities to develop and maintain natural supports, such as those accessed through social networks and community participation, play an important part in a recovery based approach [56, 57]. The recently launched National Disability Insurance Scheme in Australia [58] which provides individualised funding packages for psychosocial services, is likely to increase the trend towards support on a one-to-one basis.

Our analysis of the workforce rate in relation to service availability provided important contextualisation of service availability data. For example, while ACT had a relatively high rate of service availability in outpatient care when compared to the other Australian regions, its low workforce rate in this area suggests that its capacity to provide this type of care could in fact be less than that of the other regions. The situation is similar in non-acute health- related outpatient services, where ACT had higher service availability than Helsinki, but a smaller workforce. This information is necessary for planners in assessing actual system capacity. Comparing workforce and service availability also provides information on team size and service distribution. The distribution of staff in a higher number of smaller teams could be less cost-efficient, but on the other hand, a higher concentration of staff in a lower number of services could also mean reduced accessibility through lower spatial distribution.

A standardised terminology is needed in the study of mental health systems to address semantic ambiguity. This ambiguity extends to how workforce roles are defined, as well as to how they are named. ISCO provides occupational definitions by role, regardless of the qualifications of the person holding the role. This is problematic in mental healthcare where the same role may be held by people with different professional backgrounds, offering different skill sets and levels of expertise [59]. Additionally, there is international variation in requirements of pathways to registration of health professionals, as well as differences in scope of practice, and in the use of regulations such as "protected title": titles which by law only people registered as a particular health professional may use.

In non-health occupations, the picture is even more confusing, with some roles differing by name or role requirements on an organisation by organisation basis. Data relating to these workers, broadly classified by the World Health Organization as the "psychosocial workforce" [10] can also be less readily available, as employers are more likely to be non- public agencies such as Non-Government Organisations, who may or may not collect or release reliable workforce data. Using the DESDE-LTC, professionals are described by their qualification and skill set, so that for example, a psychologist working in a role described as a mental health worker is distinguished from a social worker in the same role, or from someone without tertiary

qualifications. However, the complexity and ambiguity of roles in social care remain a barrier to understanding the capacity of the workforce.

## Limitations

This study did not include data from services whose availability to the consumer requires a substantial out of pocket contribution. These services could be separately analysed as another layer of the healthcare system. Some data in the study was incomplete, and thus stated figures may be lower than actual figures: namely in SESLHD in health related residential care and, in Helsinki, in residential community care acute health related and social related outpatient care; in SLHD and ACT in residential community care; and in all Australian regions in acute health related outpatient care. However, all reasonable attempts were made to obtain full data. A lack of standardised workforce terminology, particularly in non-professional roles, as described above limited analysis of this important section of the workforce in all regions.

## Implications

These results provide a baseline of workforce composition, organisation and capacity at the local level, from which analyses of current and future need can be monitored. Our study shows the critical need for workforce data obtained at this level. We identified significant differences between national and regional data in several types of care: modelling based on the national level data in these cases would provide highly misleading scenarios and lead to inappropriate and potentially inequitable allocation of resources. Critical shortages in certain occupations, particularly in mental health nursing, have been identified as imminent [4] and current data is a crucial first step in monitoring the effectiveness of strategies to address these. This study has shown the relevance of workforce data in conjunction with service availability data to provide a full picture of actual availability of care. Information on team size and distribution can inform planning for accessibility and efficiency. Additionally, workforce planning in any type of service setting needs to consider the current availability and capacity of services to increase or decrease their staff levels. Aligning workforce availability and location with consumer need, and estimating future supply needs, as recommended in the Productivity Commission Report [4] requires a mapping and measure of current supply and location. A knowledge of the make-up of the workforce in terms of professional background and skill set, such as that provided in this study, is particularly relevant to an additional Report recommendation for data on work-force characteristics, particularly in relation to workforce capabilities, to inform broader mental health service reform.

There is an urgent need for more reliable and standardised data to measure the various roles in the non-registered professional workforce, particularly given its importance in expanding community care provision. The current situation does not provide adequate information about the roles, skill level and functions being performed by workers providing non-health related or psychosocial care, thus limiting the ability to identify gaps in psychosocial care.

A systematic analysis of workforce capacity is important in comparative effectiveness studies of mental health systems, providing a strategy for a detailed analysis for modeling using real data [60], such as the effect of changes to the workforce in one part of the system on the capacity of other parts of the system.

## Future steps

In addition to current workforce provision, analysis of the workforce in training, including completion rates could inform planners of future capacity. A comparison of the workforce

provision in rural areas is also needed to identify possible issues of inequity between rural and urban workforce availability.

## Conclusion

This study has shown the usability of an ecosystem approach using a standardised classification instrument in a comparison of workforce profiles and capacity in mental healthcare. We have identified patterns in care provision that reflect the whole care system and that are internationally comparable. A comparison of workforce capacity and composition is critical to provide planners with information about other workforce models and enable comparison towards specific targets. The use of meaningful information about the local area can help in understanding capacity, and contextualising better the size of care teams and professionals by service type. Role ambiguity, particularly in the non-professional and community workforce sector impedes accurate monitoring and onfounds attempts at needs based planning.

## Acknowledgments

We would like to acknowledge the PHNs, also Bruno Aloisi, Lauren Anthes, Psicost, Universidad Loyola Andalucía.

## Author Contributions

**Conceptualization:** Mary Anne Furst, Mencia R. Gutiérrez-Colosia, Luis Salvador-Carulla.

**Data curation:** Mary Anne Furst.

**Formal analysis:** Jose A. Salinas-Perez.

**Investigation:** Mary Anne Furst.

**Methodology:** Mary Anne Furst, Luis Salvador-Carulla.

**Project administration:** Mary Anne Furst.

**Supervision:** Luis Salvador-Carulla.

**Visualization:** Jose A. Salinas-Perez.

**Writing – original draft:** Mary Anne Furst.

**Writing – review & editing:** Jose A. Salinas-Perez, Mencia R. Gutiérrez-Colosia.

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
