## [Decision Letter · Decision Letter 0]

17 Jun 2021

PONE-D-21-05096

A new bottom-up method for the standard analysis and comparison of workforce capacity in mental healthcare planning: demonstration study in the Australian Capital Territory

PLOS ONE

Dear Dr. Furst,

Thank you for submitting your manuscript to PLOS ONE. After careful consideration, we feel that it has merit but does not fully meet PLOS ONE’s publication criteria as it currently stands. Therefore, we invite you to submit a revised version of the manuscript that addresses the points raised during the review process.

The Academic Editor reviewed the manuscript as a second reviewer and agreed that the following question needs to be addressed: "Are counsellors and psychotherapists a sub category of the professions you are listing? It wasn't clear why those professions are not included? For the international audience (which I am one of) possibly clarifying the inclusion of Occupational Therapists (which wouldn't necessarily be an included profession in the United States for mental health services) while not including some of the others might be helpful."

Once adequately addressed, the manuscript can move forward for consideration for publication.

We look forward to receiving your revised manuscript.

Kind regards,

Joseph Telfair, DrPH, MSW, MPH

Academic Editor

PLOS ONE

Journal Requirements:

Additional Editor Comments (if provided):

The Academic Editor served as the second reviewer and agreed to Accept

Reviewers' comments:

Reviewer's Responses to Questions

**Comments to the Author**

1. Is the manuscript technically sound, and do the data support the conclusions?

Reviewer #1: Yes

2. Has the statistical analysis been performed appropriately and rigorously? 

Reviewer #1: Yes

3. Have the authors made all data underlying the findings in their manuscript fully available?

Reviewer #1: Yes

4. Is the manuscript presented in an intelligible fashion and written in standard English?

Reviewer #1: Yes

5. Review Comments to the Author

Reviewer #1: Interesting study on resources and capacity of the mental health work force. Are counsellors and psychotherapists a sub category of the professions you are listing? It wasn't clear why those professions are not included? For the international audience (which I am one of) possibly clarifying the inclusion of Occupational Therapists (which wouldn't necessarily be an included profession in the United States for mental health services) while not including some of the others might be helpful.

6. PLOS authors have the option to publish the peer review history of their article (what does this mean?). If published, this will include your full peer review and any attached files.

Reviewer #1: No

---

## [Author Response · Author response to Decision Letter 0]

7 Jul 2021

Dear Reviewers,

Thank you very much for reviewing our paper and we very much appreciate your feedback. Counsellors and psychotherapists are indeed key professionals in mental health care delivery, and you have highlighted a need for us to clarify how we have included this group of professionals, as well as why other professionals not always associated with mental health care, such as occupational therapists, have been included in our study. We have responded to your comments by adding some additional clarification in the manuscript as described below.

Reviewer comment:

“Interesting study on resources and capacity of the mental health work force. Are counsellors and psychotherapists a sub category of the professions you are listing? It wasn't clear why those professions are not included? For the international audience (which I am one of) possibly clarifying the inclusion of Occupational Therapists (which wouldn't necessarily be an included profession in the United States for mental health services) while not including some of the others might be helpful”.

Response:

With regard to counsellors, and psychotherapists: we have found that these occupational titles, like many others in mental health, are ambiguous, and may refer to someone from a range of professional backgrounds. A counsellor may be a clinical psychologist or they may have a diploma in counselling for example; a psychotherapist may be a psychiatrist or a psychologist, depending on the jurisdiction. This ambiguity in occupational titles is a huge limitation in any assessment of workforce capacity, as different professionals bring different skills and experience, and thus capacity, to their role (defining “capacity” as we have in the paper: “The term “capacity” follows the Talent Management Model [25] in human resource management as “the knowledge and skills, qualifications and entitlement of an individual to conduct a defined set of activities in practice that defines the maximum ability that exists at present in real world conditions”. It is characterised by the “power, ability or possibility of doing something or performing”) [26]. This concept is different to “capability”, which refers to the higher level of ability that could be demonstrated under the right or ideal conditions. Capacity is also different from current performance, as it takes into account the knowledge and skill set of the individual”). Thus it is important to know the professional background of the person undertaking the role of counsellor or psychotherapist in order to understand the real capacity of the role. For this reason, when identifying the workforce in each service, and where ambiguity exists such as with “counsellor”or “psychotherapist” we also identify the individual’s relevant professional background and include them in the data according to this. So a counsellor who is a clinical psychologist for example is identified as such, and counted as a “core health” professional; while a counsellor with a diploma level qualification in counselling for example, would be included in the data as a non core-health professional; similarly a counsellor with tertiary social work qualifications would be counted as “allied health”.

With regard to occupational therapists: as the DESDE service classification system is based on a whole systems approach, we include all people providing direct care in services providing mental health support across all sectors. So although occupational therapists are less commonly employed in mental health care, where we have identified services employing them to provide direct support to a target population of people with mental health or psychosocial needs, they have been included as part of the “allied health professional” group.

To clarify these important points in the paper, we have made the following revisions to the manuscript:

1. In paragraph 3 of the introduction, we have added “and counsellors” to the occupational umbrella terms described as ambiguous, so that this sentence now reads: “Umbrella terms such as “case manager” and “counsellor” describe roles which may be occupied by any of several different types of professional, each bringing quite different skill sets”.

2. In paragraph 6, the final paragraph of the methods subsection “Key models, terms and groupings”, we have modified the description of workforce groupings to read as follows: 

“In our study, we have included all people employed to provide direct support to the target population of each service, according to their professional background. This addresses the issues of ambiguity regarding occupational titles such as “counsellor”, “psychotherapist” or “case manager” where the position may be held by a range of different professionals, and provides a more accurate picture of the real capacity of the workforce accordingly. The “clinical health professionals” group included psychiatrists/registrars, other physicians, psychologists, and nurses; while “allied health professionals” refers to any tertiary qualified allied health professional employed to provide direct care to the target population, such as social workers, or less frequently, occupational therapists”.

---

## [Decision Letter · Decision Letter 1]

15 Jul 2021

A new bottom-up method for the standard analysis and comparison of workforce capacity in mental healthcare planning: demonstration study in the Australian Capital Territory

PONE-D-21-05096R1

Dear Dr. Furst,

We’re pleased to inform you that your manuscript has been judged scientifically suitable for publication and will be formally accepted for publication once it meets all outstanding technical requirements.

Kind regards,

Joseph Telfair, DrPH, MSW, MPH

Academic Editor

PLOS ONE

Additional Editor Comments (optional):

The academic editor served as the second reviewer for this manuscript. All minor issues have been addressed. It is agreed that the manuscript should be accepted for publication pending any technical concerns.

Reviewers' comments:

Reviewer's Responses to Questions

**Comments to the Author**

1. If the authors have adequately addressed your comments raised in a previous round of review and you feel that this manuscript is now acceptable for publication, you may indicate that here to bypass the “Comments to the Author” section, enter your conflict of interest statement in the “Confidential to Editor” section, and submit your "Accept" recommendation.

Reviewer #1: All comments have been addressed

2. Is the manuscript technically sound, and do the data support the conclusions?

Reviewer #1: Yes

3. Has the statistical analysis been performed appropriately and rigorously? 

Reviewer #1: Yes

4. Have the authors made all data underlying the findings in their manuscript fully available?

Reviewer #1: Yes

5. Is the manuscript presented in an intelligible fashion and written in standard English?

Reviewer #1: Yes

6. Review Comments to the Author

Reviewer #1: Thank you for clarifying the role definitions. If there are licensing or certifications linked to the different job titles that might help with international appeal and relatability. In America our mental health services are governed by licensing which drives specific completion of a degree (Ph.D, PsyD,. SWD, MS, etc). It would be interesting for this study to be duplicated in other countries.

7. PLOS authors have the option to publish the peer review history of their article (what does this mean?). If published, this will include your full peer review and any attached files.

Reviewer #1: No

---

## [Editor Report · Acceptance letter]

19 Jul 2021

PONE-D-21-05096R1 

A new bottom-up method for the standard analysis and comparison of workforce capacity in mental healthcare planning: demonstration study in the Australian Capital Territory 

Dear Dr. Furst:

I'm pleased to inform you that your manuscript has been deemed suitable for publication in PLOS ONE. Congratulations! Your manuscript is now with our production department. 

Kind regards, 

on behalf of

Dr. Joseph Telfair 

Academic Editor

PLOS ONE